# Different Effects of Valproic Acid on *SLC12A2*, *SLC12A5* and *SLC5A8* Gene Expression in Pediatric Glioblastoma Cells as an Approach to Personalised Therapy

**DOI:** 10.3390/biomedicines10050968

**Published:** 2022-04-22

**Authors:** Eligija Damanskienė, Ingrida Balnytė, Angelija Valančiūtė, Marta Marija Alonso, Donatas Stakišaitis

**Affiliations:** 1Department of Histology and Embryology, Medical Academy, Lithuanian University of Health Sciences, 44307 Kaunas, Lithuania; eligija.damanskiene@lsmuni.lt (E.D.); ingrida.balnyte@lsmuni.lt (I.B.); angelija.valanciute@lsmuni.lt (A.V.); 2Department of Pediatrics, Clínica Universidad de Navarra, University of Navarra, 31008 Pamplona, Spain; mmalonso@unav.es; 3Laboratory of Molecular Oncology, National Cancer Institute, 08660 Vilnius, Lithuania

**Keywords:** valproic acid, pediatric glioblastoma, PBT24, SF8628, NKCC1, KCC2, SLC5A8

## Abstract

Valproic acid (VPA) is a histone deacetylase inhibitor with sex-specific immunomodulatory and anticancer effects. This study aimed to investigate the effect of 0.5 and 0.75 mM VPA on NKCC1 (*SLC12A2*), KCC2 (*SLC12A5*) and SLC5A8 (*SLC5A8*) co-transporter gene expressions in pediatric PBT24 (boy’s) and SF8628 (girl’s) glioblastoma cells. The *SLC12A2*, *SLC12A5* and *SLC5A8* RNA expressions were determined by the RT-PCR method. The *SLC12A2* and *SLC5A8* expressions did not differ between the PBT24 and SF8628 controls. The *SLC12A5* expression in the PBT24 control was significantly higher than in the SF8628 control. VPA treatment significantly increased the expression of *SLC12A2* in PBT24 but did not affect SF8628 cells. VPA increased the *SLC12A5* expression in PBT24 and SF8628 cells. The *SLC12A5* expression of the PBT24-treated cells was significantly higher than in corresponding SF8628 groups. Both VPA doses increased the *SLC5A8* expression in PBT24 and SF8628 cells, but the expression was significantly higher in the PBT24-treated, compared to the respective SF8628 groups. The *SLC5A8* expression in PBT24-treated cells was 10-fold higher than in SF8628 cells. The distinct effects of VPA on the expression of *SLC12A2*, *SLC12A5* and *SLC5A8* in PBT24 and SF8628 glioblastoma cells suggest differences in tumor cell biology that may be gender-related.

## 1. Introduction

Pediatric high-grade glioblastoma multiforme (phGBM) is a highly malignant brain tumor that is the leading cause of death in the pediatric brain cancer population [1,2]. Due to the difference in biological factors from adult glioblastoma and the uniqueness of this pediatric heterogeneous group of the tumor, reasonable therapeutic strategies for adult glioblastoma have not improved the outcome of phGBM treatment [3,4]. To effectively treat phGBM, it is essential to determine the individual sensitivity of the drug to carcinogenesis. The development of safe and effective anticancer medicines aims to improve the bioavailability of medicines and reduce the development of resistance to chemotherapy [5]. Equally important is the identification of the factors that make treatment effective. Such determinants may be gender-specific, which could be key to developing individualised treatments [6].

Valproic acid (2-n-propyl-pentanoic acid; VPA) is an investigational preparation for glioma treatment [7]. VPA is a well-known histone deacetylase inhibitor (HDACi) [8,9]. In combination with radiotherapy or chemotherapy, VPA delays the development of glioblastoma resistance to chemotherapy or radiotherapy [10,11,12,13,14]; in combination with radiotherapy, it increases the sensitivity of primary glioma cells to radiotherapy and γ-radiation-induced cytotoxicity [14,15]. Anti-tumor effects of VPA in combination with chemotherapy and radiotherapy are of interest for phGBM [16,17].

The metabolism of VPA is sex-specific. VPA is transported more via the G protein (MRP) in females than in males of experimental animals and humans [18,19]. Testosterone reduces drug transport [20]. Hepatic metabolism of VPA is more potent in females [21]; in males, due to lower expression of efflux pumps, a saturation of hepatobiliary transport results in longer retention of VPA in the hepatocyte, resulting in slower clearance of VPA [22].

One of the main anticancer effects of drugs is to inhibit tumor cell proliferation and induce apoptosis. VPA inhibits glioma cell proliferation and induces apoptosis [23]. The chloride anion (Cl^−^) is involved in cell volume regulation and cell apoptosis [24]. An early sign of apoptosis is a decrease in cell volume due to loss of intracellular K^+^ and chloride ([Cl^−^]i) [25,26]. High-grade glioblastoma cells are associated with increased [Cl^−^]i [27], which is associated with increased Na-K-2Cl co-transporter (NKCC1) and decreased K-Cl co-transporter (KCC2) activity [28,29]. The sodium and Cl^−^ ion-dependent SLC5A8 co-transporter triggers cell apoptosis via pyruvate-dependent HDAC inhibition [30]. In elucidating the pathogenesis of glioblastoma, it is essential to investigate the mechanisms involved in the regulation of [Cl^−^]i concerning the efficacy of therapy. The mechanisms by which VPA is involved in the regulation of [Cl^−^]i are illustrated in Figure 1.

Increased expression of the NKCC1 in glioblastoma cells is linked to cell proliferation [31]. Increased NKCC1 protein expression in human glioblastoma is directly related to tumor grade and cell migration; inhibition of NKCC1 attenuates glioblastoma cell migration and tumor invasion [27,32,33]. VPA suppresses NKCC1 gene expression in male, but not female, rat thymocytes [34]. Reduction of KCC2 neuropil staining was reported in adult patients with glioblastoma and epilepsy [29]. VPA produces both kaliuretic and chloriduretic effects in males but not in female rats [35]. Whether the VPA initiates KCC2 activity in pediatric glioblastoma cells, leading to a parallel loss of K^+^ and Cl^−^ ions, needs to be investigated.

SLC5A8 is a sodium (Na^+^) and Cl^−^ dependent and sodium-linked monocarboxylate co-transporter that transports lactate, pyruvate, acetate, propionate, valerate, butyrate, and monocarboxylate preparations (dichloroacetate, nicotinate, salicylate) into cells [36,37]. SLC5A8 triggers cell apoptosis through pyruvate-dependent HDAC inhibition [30]. *SLC5A8* is a tumor growth-inhibitory gene in primary human and experimental animal glioma and is suppressed by epigenetic mechanisms [35]. The epigenetic reprogramming of cancer cells induced by VPA is dependent on estrogen receptors [38]. Suppression of the *SLC5A8* gene is linked to DNA methylation, and treatment of cancer cells with DNA demethylating agents increases *SLC5A8* expression [37]. VPA may also activate genes regulated by DNA methylation, and the VPA treatment may be triggered by active DNA demethylation in cancer cells [39,40]. *SLC5A8* activation in male rat duodenal enterocytes is correlated with NKCC1 activity [41]. In fibroblasts treated with the NKCC1 inhibitor bumetanide, an enhancement of maximal respiration indicates an increase in substrate availability and mitochondrial oxidation [42], indicating that altered NKCC1 activity may result in alterations in mitochondrial function.

The study aimed to investigate if there are differences in VPA’s effect on the expression of NKCC1, KCC2 and SLC5A8 co-transporter genes between high-grade SF8628 cells and high-grade PBT24 cells.

## 2. Materials and Methods

### 2.1. Cell Lines and Cell Culture

A 13-year-old boy’s high-grade glioblastoma PBT24 cell line cells were donated by Prof. M. M. Alonso (University of Navarra, Spain) [43] for the study. A 3-year-old girl’s diffuse intrinsic pontine glioblastoma (DIPG) SF8628 cell line cells—harboring the histone H3.3 Lys 27-to-methionine (Sigma Aldrich, St. Louis, MO, USA)—were also studied [44,45]. The PBT24 cells were cultivated in Roswell Park Memorial Institute 1640 (RPMI), medium (Sigma Aldrich, St. Louis, MO, USA) supplemented with 10% fetal bovine serum (FBS; Sigma Aldrich, St. Louis, MO, USA) containing 100 IU/mL of penicillin and 100 µg/mL of streptomycin (P/S; Sigma Aldrich, St. Louis, MO, USA). The SF8628 cells were cultivated in Dulbecco’s Modified Eagle Medium (DMEM)–High Glucose Media (Sigma Aldrich, St. Louis, MO, USA), supplemented with 10% fetal bovine serum (FBS; Sigma Aldrich, St. Louis, MO, USA) containing 100 IU/mL of penicillin and 100 µg/mL of streptomycin (P/S; Sigma Aldrich, St. Louis, MO, USA) and 2 mM L-Glutamine (Sigma Aldrich, St. Louis, MO, USA). Cells were incubated at 37 °C in a humidified 5% CO_2_ atmosphere.

### 2.2. Extraction of RNA from PBT24 and SF8628 Cell Line Cells

PBT24 and SF8628 cell line cells were treated with 0.5 mM VPA and 0.75 mM VPA for 24 h. The concentration of 0.5 mM VPA and 0.75 mM VPA was chosen because it corresponds to the mean plasma concentration of the drug in VPA-treated patients [46,47]. Control groups were cultured in a cell culture medium depending on the cell line. According to the manufacturer’s instructions, the total RNA was extracted using the TRIzol Plus RNA Purification Kit (Life Technologies, New York, NY, USA). The RNA quality and concentration were assessed using a NanoDrop2000 spectrophotometer (Thermo Scientific, Branchburg, NJ, USA). The total RNA integrity was analyzed using the Agilent 2100 Bioanalyzer system (Agilent Technologies, Santa Clara, CA, USA) with an Agilent RNA 6000 Nano Kit (Agilent Technologies, Santa Clara, CA, USA). The samples of RNA were stored at −80 °C until further analysis.

### 2.3. Determination of the SLC12A2, SLC12A5 and SLC5A8 Expression in PBT24 and SF8628 Cell Line Cells

RNA expression assays were performed for *SLC12A2* (Hs00169032_m1), *SLC12A5* (Hs00221168_m1), *SLC5A8* (Hs00377618_m1) and *GAPDH* (Hs02786624_g1) genes. According to the manufacturer’s instructions, reverse transcription was performed with the High-Capacity cDNA Reverse Transcription Kit with RNase Inhibitor (Applied Biosystems, Waltham, MA, USA) in a 20 µL reaction volume containing 50 ng RNA using the Biometra TAdvanced thermal cycler (Analytik Jena AG, Jena, Germany). The synthesized copy DNA (cDNA) was stored at 4 °C until use, or at −80 °C for a longer time. Real-time polymerase chain reaction (PCR) was performed using an Applied Biosystems 7900 Fast Real-Time PCR System (Applied Bio-systems, Waltham, MA, USA). The reactions were run in triplicate with 4 µL of cDNA template in a 20 µL reaction volume, 10 µL of TaqMan Universal Master Mix II, no UNG (Applied Biosystems, Waltham, MA, USA), 1 µL of TaqMan Gene Expression Assay 20× (Applied Biosystems, Waltham, MA, USA), 5 µL of nuclease-free water (Invitrogen, Carlsbad, CA, USA), with the program running at 95 °C for 10 min, followed by 45 cycles of 95 °C for 15 s, 50 °C for 2 min and 60 °C for 1 min.

The control and 24 h VPA-treated groups (n = 6 per group) were tested for *SLC12A2*, *SLC12A5*, *SLC5A8* and *GAPDH* expression.

### 2.4. Statistical Analysis

To investigate the *SLC12A2*, *SLC12A5*, and *SLC5A8* RNA gene expression in the VPA-treated and control groups, the threshold cycle (CT) value was normalized with the control *GAPDH*, and the ΔCT value was obtained. The Livak method (∆∆CT) was used for calculating the relative fold change in expression levels [48]. The Spearman’s rank correlation coefficient (*r*) was used to assess relationships between the *SLC12A2* and *SLC12A5* (ΔCT values were used). Differences at the value of *p* < 0.05 were considered significant. The figures were created using GraphPad Prism 7 and IBM SPSS Statistics 23.0 software.

## 3. Results

### 3.1. The Expression of SLC12A2 in PBT24 and SF8628 Cell Study Groups

The *SLC12A2* expression data for the PBT24 and SF8628 cell groups tested are shown in Table 1 and Figure 2.

A comparison of the *SLC12A2* expression of PBT24 and SF8628 cells between the control groups showed no difference. Treatment with the tested doses of VPA significantly increased the *SLC12A2* expression of PBT24 cells. No significant VPA effect on SF8628 cells was observed. After 24 h of treatment with the tested doses of VPA, the *SLC12A2* expression of SF8628 cells was significantly lower than that of PBT24 cells treated with 0.5 and 0.75 mM of VPA.

Compared to the PBT24 control, the 0.5 mM VPA dose increased *SLC12A2* expression in PBT24 cells 1.9-fold (2^−ΔΔCT^ = 1.858), while the 0.75 mM VPA dose increased *SLC12A2* expression in PBT24 cells 2.1-fold without significant difference between the two treated groups (Figure 2B).

### 3.2. The Expression of SLC12A5 in PBT24 and SF8628 Cell Study Groups

The expression data of *SLC12A5* and *GAPDH* in PBT24 and SF8628 cell controls and in the groups treated with two doses of VPA for 24 h are shown in Table 2.

The expression of *SLC12A5* in PBT24 control cells is significantly higher than that of SF8628 control cells. Comparison of PBT24 control and 0.5 mM VPA-treated cells showed substantially higher *SLC12A5* expression in treated cells. Comparing the expression of SLC12A5 in PBT24 control and 0.75 mM VPA-treated cells, there is a clear trend towards an increase in expression in treated cells, but no statistical difference was found. No difference in the *SLC12A5* expression was determined when comparing 0.5 mM and 0.75 mM PBT24 treated cells. The *SLC12A5* expression in PBT24 cells treated with the tested doses of VPA was significantly higher than in the corresponding SF8628 cell groups.

When comparing the expression of *SLC12A5* in SF8628 control with those treated with a 0.75 mM dose of VPA, the expression was significantly higher in treated cells. Compared to control, no statistically significant change in *SLC12A5* expression was observed in dose cells treated with 0.5 mM VPA (Table 2; Figure 3A). There was no significant difference in *SLC12A5* expression between SF8628 cells treated with different VPA doses (*p* > 0.05).

Compared to the respective control, treatment with 0.5 mM VPA dose increased *SLC12A5* expression 2.2-fold (2^−ΔΔCT^ = 2.220) in PBT24 cells and 1.5-fold in SF8628 cells; treatment with 0.75 mM VPA increased the expression of *SLC12A5* in PBT24 cells 2-fold and that of SF8628 cells 1.8-fold (Figure 3B).

Comparing the gene expression of the tested cell controls, PBT24 cells’ *SLC12A5* was expressed 24.6-fold (2^−ΔΔCT^ = 24.625) over SF8628 cells. The *SLC12A5* expression of PBT24 cells treated with 0.5 mM VPA was 37-fold (2^−ΔΔCT^) higher over SF8626-0.5 mM VPA cells; those treated with 0.75 mM VPA were 28-fold higher in PBT24 cells than in SF8628 cells.

The correlation (*r*) between ΔCT *SLC12A2* and ΔCT *SLC12A5* values of the study groups was the following: 0.71 for the PBT24 cell control (*p* < 0.05), 0.14 for the SF8628 cell control (*p* > 0.05), 0.03 for the PBT24 cells treated with 0.5 mM VPA (*p* > 0.05) and 0.37 for the SF8628 cells treated with 0.5 mM VPA (*p* > 0.05; Figure 4); 0.37 for the PBT24 cells treated with 0.75 mM VPA, and −0.08 for the SF8628 cells treated with 0.75 mM VPA (*p* > 0.05 in groups). VPA treatment significantly altered the correlation between *SLC12A2* and *SLC12A5* in the PBT24 cell. These changes suggest that VPA may reduce the [Cl^−^]i of the PBT24 cell through activation of *SLC12A5* and down-regulation of *SLC12A2*. In contrast, we did not find a trend towards such an effect in the SF8628 cells.

### 3.3. The Expression of SLC5A8 in PBT24 and SF8628 Cell Study Groups

The expression data for *SLC5A8* and *GAPDH* in PBT24 and SF8628 cells in the control and 0.5 and 0.75 mM VPA-treated groups for 24 h are shown in Table 3.

The expression of *SLC5A8* in PBT24 control cells was not different from that in SF8628 control. *SLC5A8* expression was significantly higher in the PBT24 cell groups treated with both VPA doses than in the corresponding groups of SF8628 cells. Compared with the respective control, treatment with VPA doses significantly increased *SLC5A8* expression in PBT24 and SF8628 cells (Table 3; Figure 5A).

Compared to the respective control, treatment with 0.5 mM VPA dose increased the *SLC5A8* expression 22.9-fold (2^−ΔΔCT^ = 22.907) in PBT24 cells and 2-fold in SF8628 cells; treatment with 0.75 mM VPA increased the *SLC5A8* expression in PBT24 cells by 37.3-fold and by 3-fold in the SF8628 cells (Figure 5B).

The *SLC5A8* expression in PBT24 cells treated with 0.5 mM VPA was 10.5-fold (2^−ΔΔCT^ = 10.475) higher than that in SF8628 cells, while *SLC5A8* expression in PBT24 cells treated with 0.75 mM VPA was 11.4-fold more heightened than in SF8628 cells.

## 4. Discussion

The efficacy of treatment of glioblastoma in the presence of high tumor polymorphism suggests that a single strategy to achieve an effective treatment is unwise [49]. The goal should be the personalised treatment of pediatric glioblastoma, tailored to the individual patient’s sex-specific molecular and genetic characteristics of tumor carcinogenesis [4].

Recently reported experimental in vivo data have shown the different effects of temozolomide (TMZ) on PBT24 and SF6828 xenografts and the molecular mechanisms involved. TMZ reduced PBT24 tumor growth, and treatment efficacy was related to significantly increased *SLC12A5* expression in PBT24 cells, but there were no such effects on *SLC12A5* expression of SF8628 cells [50].

The [Cl^−^]i content of high-grade glioblastoma cells is significantly higher than the average in grade II glioma and normal cortical cells [27]. Increased [Cl^−^]i level in glioblastoma cells are associated with increased NKCC1 and decreased K-Cl co-transporter activity [28,29]. Na-K-2Cl co-transporter activity in glioblastoma is associated with cell proliferation [31]; the increased expression of NKCC1 protein in human glioblastoma is associated with tumor grade; inhibition of NKCC1 activity impairs tumor invasion [27,32]. Strong NKCC1 immune reactivity is characteristic of the aberrant neuronal component of the glioblastoma, while the increase in neuronal NKCC1 expression is not characteristic of the peri-lesional zone of tumor samples. The high levels of neuronal NKCC1 in phGBM support the hypothesis of abnormal and immature neuronal cells [29]. The study showed no difference in *SLC12A2* expression between PBT24 and SF8628 cell controls, but VPA treatment significantly increased the expression of *SLC12A2* in PBT24 cells, while no such effect was determined in SF8628 cells. The mechanism of this VPA action is unknown. TMZ treatment increased NKCC1 expression in PBT24 and SF8626 cells [50]. TMZ-induced NKCC1 activation is associated with increased WNK kinases protein phosphorylation [51,52,53]. The suppressive effect on glioma cell [Cl^−^]i may be due to inhibition of NKCC1 activity and other Cl^−^ transport mechanisms involved in Cl^−^ efflux from the cell [33,50].

Our study shows that the PBT24 control cell *SLC12A5* expression is significantly higher than the SF8628 controls. When comparing the *SLC12A5* expression between the PBT24 and the SF8628 controls and cells treated with VPA, the expression was significantly higher in both lines of treated cells, while the expression of *SLC12A5* was explicitly higher in VPA-treated PBT24 cells than in the corresponding SF8628 groups. It cannot be ruled out that such differences between study cells are also related to gender.

A key feature of cancer cell apoptosis is a reduction in cell volume due to a decrease in [K^+^]i and [Cl^−^]i [25,26]. Apoptosis requires a steady decrease in cell volume due to the loss of [K^+^]i and [Cl^−^]i, which occurs before other detectable signs of apoptosis [24,54,55]. The decrease in [K^+^]i and [Cl^−^]i is associated with caspase activation and caspase cascade-related apoptotic mechanisms [26]. Furthermore, KCC2 is a neuron-specific K^+^ and Cl^−^ extruder that uses a K^+^ gradient to maintain low [Cl^−^]i levels and ensure the proper functioning of postsynaptic GABAA receptors. The paradoxical excitatory effects of GABAA depend on the relatively high [Cl^−^]i content [56]. Increased activity of NKCC1 can lead to astrocyte swelling [57,58] and induce a GABAA receptor-mediated excitatory response that promotes seizures [56,58,59]. Reduced KCC2 neuropil staining has been found in glioblastoma of patients with marked epilepsy [29]. Reduced KCC2 expression and function is a hallmark of evolving cerebral epileptic disorders [60]. The effect of drugs that activate KCC2 function in glioblastoma cells is of interest as a potential new anticancer therapeutic target. Future research on the co-localization of Cl^−^ co-transporters with the GABAA receptor may shed light on the importance of the functional interaction of Cl^−^ transporters in glioblastoma cells.

The VPA-associated anti-tumor activity has been observed in children with refractory and those with phGBM who received VPA monotherapy for progressive phGBM after radiotherapy and chemotherapy [16,61,62,63]. However, a meta-analysis of 1869 patients enrolled in five phase III trials for newly diagnosed glioblastoma found no improvement with the addition of VPA to standard chemotherapy [64]. Such conflicting data on VPA efficacy suggest that future studies could differentiate treatment efficacy by individual characteristics. An example of a potential non-effective treatment or induced tumor progression could be the combination of VPA with temozolomide—a case where both preparations would activate NKCC1 function synergistically.

Another important cancer marker in determining the effectiveness of VPA could be the SLC5A8 co-transporter, which acts as an electrogenic sodium- and chloride-dependent sodium-coupled co-transporter [65,66]. The *SLC5A8* acts as a tumor growth suppressor gene, often silenced by epigenetic mechanisms in primary gliomas [35]. Silencing the *SLC5A8* is associated with DNA methylation, and treatment of cancer cells with DNA demethylating agents increases the *SLC5A8* expression [37]. VPA inhibits HDAC activity [67], inducing DNA demethylation and increasing the *SLC5A8* expression [37,39,40]. *SLC5A8* induces cell apoptosis via pyruvate-dependent inhibition of HDAC [30]. Our data show that the expression of *SLC5A8* in control PBT24 cells did not differ from that of SF8628 controls. Compared to the respective control, treatment with both doses of VPA significantly increased *SLC5A8* expression in PBT24 and SF8628 cells, but *SLC5A8* expression was explicitly higher in VPA-treated PBT24 cells. These data show that VPA has a stronger tumor-suppressive effect on PBT24 cells (boy’s) than SF8626 (girl’s) cells. The *SLC5A8* activity in the male rat duodenum enterocytes depends on Na^+^-K^+^-2Cl^−^ cotransporter (NKCC1) activity [41]. Changes in the NKCC1 activity might lead to mitochondrial SLC5A8-related function changes [42]. The differences in VPA effects found between PBT24 and SF8628 pediatric glioblastoma may indicate medicine differential effects on carcinogenesis linked to [Cl^−^]i regulation, as PBT24 cells showed a higher expression of the KCC2 gene as well as a more pronounced activation of the co-transporter gene under VPA exposure. The effect of VPA on KCC2 is inseparable from the activity of the GABAA receptor. The role of VPA treatment on NKCC1, KCC2, and GABAA in the regulation of [Cl^−^]i is an important point for further investigation. VPA also differently activated the expression of the *SLC5A8* gene in the study cells. Furthermore, it cannot be excluded that the found differences between PBT24 and SF8628 are related to sex. Thus, assessing the gender of the cell is key to determining the efficacy of anticancer drugs, as it allows the molecular subtype of glioblastoma to be identified and allows for more effective personalised therapies to be developed based on molecular mechanisms that are also linked to sex [68].

The detected differences in VPA treatment between the compared glioblastoma cells may be therapeutically significant. The several-fold increase in the expression of *SLC12A5* and *SLC5A8* after VPA treatment directly reflects correlation with the RNA gene methylation. The mRNA assay is used to quantify gene expression and is a more accurate identifier of gene activity than the DNA methylation test [69]. According to the scientific literature, the correlation between differentially expressed mRNA and mRNA/protein of the same gene is debatable. Genome-wide correlations between mRNA and protein are low [70,71,72]. Thus, the limitation of the protein expression investigation of the studied genes can also be justified by the doubtful likelihood that short-term treatment with VPA would be associated with changes in protein expression. Therefore, evidence of apparent differences in gene expression obtained after VPA treatment is significant. A limitation of our study is that other mechanisms underlying protein activity were not excluded. The study data include NKCC1, which determines Cl^−^ influx, and KCC2, which resolves anion efflux in a cell. Therefore, it would be further meaningful to investigate the VPA effect on [Cl^−^]i, which reflects Cl- transport mechanisms and is a signaling pathway. Such follow-up research would also allow assessment of the effect of VPA on the functioning of the Cl^−^ anion-dependent SLC5A8 co-transporter.

## 5. Conclusions

The different effects of VPA on PBT24 and SF8628 glioblastoma cells are associated with a higher impact of VPA on SLC12A5, which may be related to the more effective reduction of [Cl^−^]i level in PBT24 cells. VPA more strongly promotes *SLC5A8* activation in PBT24 cells. The different effects of VPA on the expression of *SLC12A2*, *SLC12A5,* and *SLC5A8* in PBT24 and SF8628 glioblastoma cells suggest differences in tumor cell biology. The study shows the importance of investigating the mechanisms of chloride anion transport across the cell membrane and their possible relationship with VPA treatment. Further research is needed on the association of VPA and gender with the treatment effectiveness on glioma.

## Figures and Tables

**Figure 1 biomedicines-10-00968-f001:**
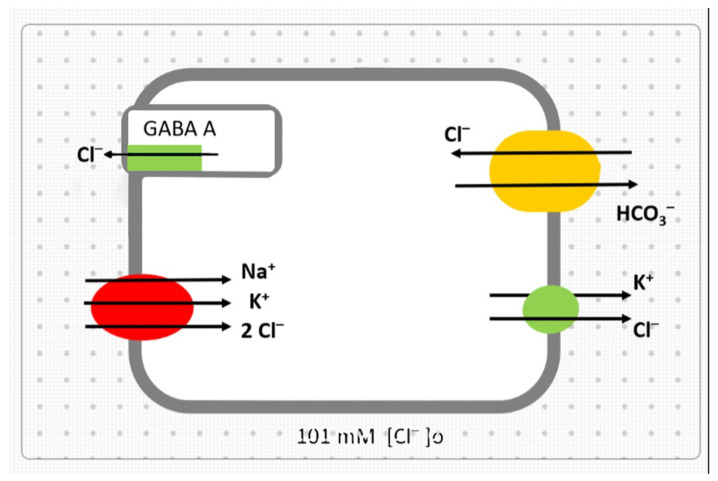
Mechanisms of Cl^−^ transport in glioblastoma cell and possible effects of VPA. The Na^+^-K^+^-2Cl^−^ co-transporter activity (Cl^−^ influx) is elevated in the glioblastoma cell, and the drug should inhibit this transport. The activity of the K^+^-Cl^−^ co-transporter (Cl^−^ efflux) is blunted (the target effect of the treatment is to activate the carrier). The activity of the latter carrier stimulates the activity of the GABA A receptor, which is a Cl^−^ channel transporting Cl^−^ out of the cell. As the concentration of [Cl^−^]i in the glioblastoma cell is significantly increased, the resulting Cl^−^ concentration gradient makes the Cl^−^/HCO_3_^−^ exchanger inactive. It is expected that the VPA can perform the necessary regulation of Cl^−^ transport across cell membrane according to the markings in figure traffic light colors.

**Figure 2 biomedicines-10-00968-f002:**
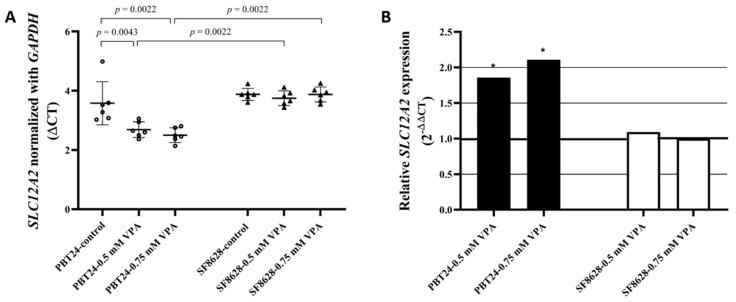
(**A**) *SLC12A2* expression in PBT24 and SF8628 control groups and VPA-treated groups. Data are after normalization with *GAPDH*. Delta threshold cycle (ΔCT) values were used for the graph (the horizontal bars represent the mean; the short horizontal lines show standard deviation (SD) values). (**B**) *SLC12A2* relative expression in PBT24 and SF8628 VPA-treated groups compared with respective control. The 1.0 line shows the starting point of gene expression; * *p* < 0.05.

**Figure 3 biomedicines-10-00968-f003:**
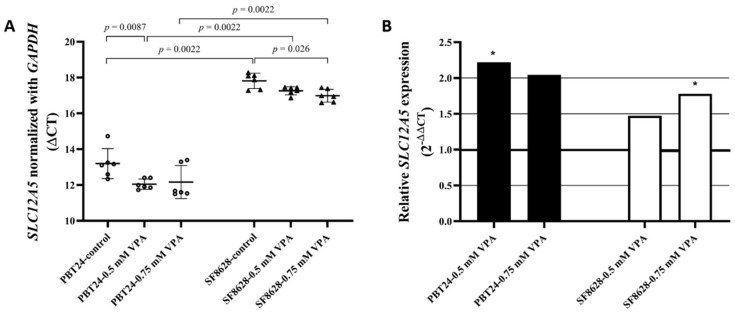
(**A**) *SLC12A5* expression in PBT24 and SF8628 control groups and VPA-treated groups. Data are after normalization with *GAPDH*. Delta threshold cycle (ΔCT) values were used for the graph (the horizontal bars represent the mean; the short horizontal lines show SD values). (**B**) The *SLC12A5* relative expression in PBT24 and SF8628 VPA-treated groups compared with respective control. The 1.0 line shows the starting point of gene expression; * *p* < 0.05.

**Figure 4 biomedicines-10-00968-f004:**
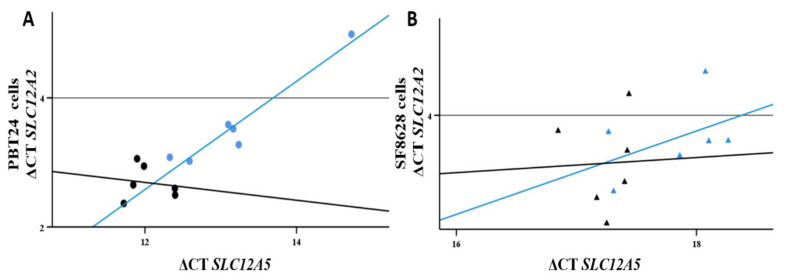
Correlation plots of the *SLC12A2* and *SLC12A5* in the PBT24 (**A**) and SF8628 (**B**) cell study groups. Blue color represents control group, black—0.5 mM VPA-treated group.

**Figure 5 biomedicines-10-00968-f005:**
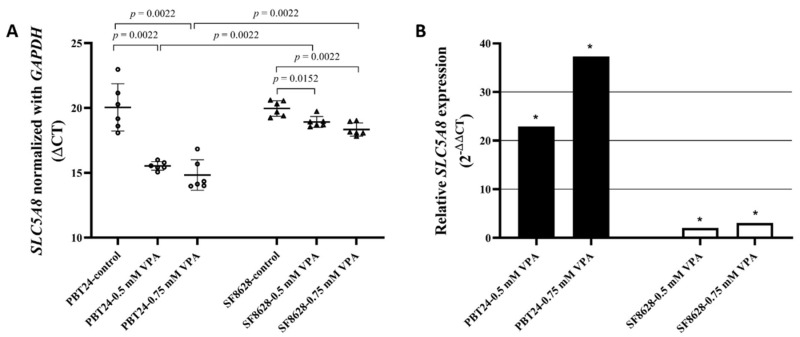
(**A**) *SLC5A8* expression in PBT24 and SF8628 control groups and VPA-treated groups. Data are after normalization with *GAPDH*. Delta threshold cycle (ΔCT) values were used for the graph (the horizontal bars represent the mean; the short horizontal lines show SD values). (**B**) *SLC5A8* relative expression in PBT24 and SF8628 VPA-treated groups compared with respective control. The 1.0 line shows the starting point of gene expression; * *p* < 0.05.

**Table 1 biomedicines-10-00968-t001:** RNA expression of *SLC12A2* in PBT24 and SF8628 cell study groups.

Study Group	n	CT Mean		ΔCT Mean ± SD	ΔΔCT
		*SLC12A2*	*GAPDH*		
PBT24-control	6	22.951	19.372	3.579 ± 0.73	
0.5-mM VPA	6	21.934	19.249	2.685 ± 0.27 ^a^	−0.894
0.75-mM VPA	6	21.847	19.346	2.501 ± 0.25 ^b^	−1.077
SF8628-control	6	22.894	19.017	3.876 ± 0.21	
0.5-mM VPA	6	23.340	19.592	3.748 ± 0.25 ^c^	−0.128
0.75-mM VPA	6	23.584	19.709	3.875 ± 0.25 ^d^	−0.002

^a^ *p* = 0.0043, compared with PBT24-control; ^b^
*p* = 0.0022, compared with PBT24-control; ^c^
*p* = 0.0022, compared with PBT24-0.5 mM VPA; ^d^
*p* = 0.0022, compared with PBT24-0.75 mM VPA.

**Table 2 biomedicines-10-00968-t002:** RNA expression of *SLC12A5* in PBT24 and SF8628 cell study groups.

Study Group	n	CT Mean		ΔCT Mean ± SD	ΔΔCT
		*SLC12A5*	*GAPDH*		
PBT24-control	6	32.564	19.372	13.191 ± 0.83	
0.5-mM VPA	6	31.290	19.249	12.041 ± 0.29 ^a^	−1.150
0.75-mM VPA	6	31.505	19.346	12.159 ± 0.92	−1.032
SF8628-control	6	36.831	19.017	17.814 ± 0.43 ^b^	
0.5-mM VPA	6	36.848	19.592	17.256 ± 0.23 ^c^	−0.558
0.75-mM VPA	6	36.691	19.709	16.982 ± 0.36 ^d,e^	−0.832

^a^*p* = 0.0087, compared with PBT24-control; ^b^
*p* = 0.0022, compared with PBT24-control; ^c^
*p* = 0.0022, compared with PBT24-0.5 mM VPA; ^d^
*p* = 0.0260, compared with SF8628-control; ^e^
*p* = 0.0022, compared with PBT24-0.75 mM VPA.

**Table 3 biomedicines-10-00968-t003:** RNA expression of *SLC5A8* in PBT24 and SF8628 cell study groups.

Study Group	n	CT Mean		ΔCT Mean ± SD	ΔΔCT
		*SLC5A8*	*GAPDH*		
PBT24-control	6	39.416	19.372	20.044 ± 1.82	
0.5-mM VPA	6	34.775	19.249	15.526 ± 0.33 ^a^	−4.518
0.75-mM VPA	6	34.170	19.346	14.824 ± 1.17 ^b^	−5.220
SF8628-control	6	38.972	19.017	19.955 ± 0.59	
0.5-mM VPA	6	38.507	19.592	18.915 ± 0.42 ^c,d^	−1.040
0.75-mM VPA	6	38.044	19.709	18.335 ± 0.51 ^e,f^	−1.620

^a^*p* = 0.0022, compared with PBT24-control; ^b^
*p* = 0.0022, compared with PBT24-control; ^c^
*p* = 0.0152, compared with SF8628-control; ^d^
*p* = 0.0022, compared with PBT24-0.5 mM VPA; ^e^
*p* = 0.0022, compared with SF8628-control; ^f^
*p* = 0.0022, compared with PBT24-0.75 mM VPA.

## Data Availability

The data presented in this study are available on request from the corresponding author.

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
