# Peer review of "Different Effects of Valproic Acid on SLC12A2, SLC12A5 and SLC5A8 Gene Expression in Pediatric Glioblastoma Cells as an Approach to Personalised Therapy"

_biomedicines, 2022, doi:10.3390/biomedicines10050968_

Round 1
Reviewer 1 Report
The manuscript, “Different Effects of Valproic Acid on Pediatric Glioblastoma PBT24 and SF8628 Cells as a Key to Personalisation of Therapy” by Damanskiene and colleagues reports the changes in expression of 3 co-transporters in male and female glioblastoma cell lines following treatment with Valproic acid (VPA). Using qRT-PCR assays, the authors document differential effects of VPA treatment on the expression of these co-transporters on male vs female cells. While these make for interesting preliminary observations, this manuscript is not ready for publication in its current form because of the extremely limited data set, and the lack of mechanistic experiments to support the authors’ inferences. These and other concerns are elaborated below:
- The authors have used only one cell line for each gender and this is not sufficient to draw any kind of conclusions. To support their observations in these cell lines and to determine the generalizability, the authors should look at patient-derived cells (from male and female patients) as well as multiple cell lines of each gender. Data derived from one cell line each is simply not enough to make generalizable conclusions.
- The only kind of data presented here is qRT-PCR. To confirm these results as well as to investigate the possible explanations that the authors come up with for their observations, protein expression (western blot) as well as functional assays, ion channel/intracellular ion concentration determination assays and exploration of possible signaling pathways would be needed, none of which has been done.
- In the introduction, the authors say that VPA suppressed NKCC1 expression in thymocytes, but it seems to increase NKCC1 expression in GBM cells. What is the (possible) explanation for this?
Author Response
Response to 1st Reviewer
|
X. English language and style are fine/minor spell check required |
Answer
English was corrected.
Is the research design appropriate?
(x) Must be improved
Answer
In response to the comment, we provide below a Figure to justify our study design.
The Figure shows the mechanisms regulating Cl- transport and intracellular Cl- concentration in a GBM cell.
GBM cells are characterized by increased expression and activity of the Na-K-2Cl- co-transporter (NKCC1), which leads to increased intracellular Cl- concentration (the aim of the treatment would be to inhibit NKCC1), and by an impaired function of the K-Cl- cotarnsporter (KCC2), which is responsible for the transport of Cl- out of the cell (the aim of treatment would be to activate it). The Cl-/HCO3 exchanger, which is not functional in GBM cells, is due to a gradient between [Cl-]i and extracellular Cl- concentration. The K-Cl- co-transporter is directly linked to the GABA A receptor's function, which is a Cl- channel (Cl- efflux of the cell). The design of the study is based on literature data (presented in the "Introduction" and "Discussion"). It is also very important to highlight that NKCC1 activity is linked to the SLC5A8 co-transporter function (the aim of treatment would be to activate SLC5A8), which is an important tumor suppressor. It is important to note that [Cl-]i level and the expression of the associated co-transporters are themselves signalling factors regulating cell proliferation and apoptosis pathways. Therefore, we believe that the study design is valid and original.
We have placed the captioned Figure in the Introduction. We believe this helps to clarify the study design. Thank you very much for your comment.
Are the conclusions supported by the results?
(x)Must be improved
Answer
Conclusions were corrected. Thanks reviewer for comment.
The manuscript, “Different Effects of Valproic Acid on Pediatric Glioblastoma PBT24 and SF8628 Cells as a Key to Personalisation of Therapy” by Damanskiene and colleagues reports the changes in expression of 3 co-transporters in male and female glioblastoma cell lines following treatment with Valproic acid (VPA). Using qRT-PCR assays, the authors document differential effects of VPA treatment on the expression of these co-transporters on male vs female cells. While these make for interesting preliminary observations, this manuscript is not ready for publication in its current form because of the extremely limited data set, and the lack of mechanistic experiments to support the authors’ inferences. These and other concerns are elaborated below:
- The authors have used only one cell line for each gender and this is not sufficient to draw any kind of conclusions. To support their observations in these cell lines and to determine the generalizability, the authors should look at patient-derived cells (from male and female patients) as well as multiple cell lines of each gender. Data derived from one cell line each is simply not enough to make generalizable conclusions.
Answer
Thanks to Reviewer for his comment. Two cell lines were used for the study and the results of the two GB cell lines were compared. The data show that there is a clear difference in the effect of VPA between the cells tested. We indicate that the compared cells are of different sexes, these indicates that the relationship of the differences found cannot be excluded to be sex-related. We agree that the conclusions are subject to adjustment and we have done so.
Regarding the generalization of the data, we would like to point out that the principle of personalized treatment should not be guided by generalization approach, as the data on the effects of the drug are applied on an individual basis. A comparison between two cells does not contradict this principle. We agree that further studies are needed to assess the VPA efficacy data found.
Also important is the reviewer's comment that it is not sufficient to perform studies only on cell lines cells, but it is essential to perform them on primary patient-derived cells (from male and female patients). We are currently conducting such studies and have tested a few of adult male and female post-operative GBM cells for the expression of NKCC1, KCC2, SLC5A8 gene co-transporters. We are comparing the effect of the drug on the tumor growth on chicken chorioallantoic membrane (CAM) model, tumor invasion into CAM and histological expression of biomarkers in tumor formed by patient-derived cells on CAM. Those data will be published, but we would like to reiterate that the expression of NKCC1, KCC2, SLC5A8 genes significantly correlates with tumor growth inhibition. Therefore, we believe that the data presented in the manuscript obtained with cell lines cells are important and allow us to perform studies with patient-derived cells, assessing the characteristics of tumor cells harvested after GBM surgery. Such data may allow us to consider whether it is appropriate to prescribe a medicine to treat an individual patient having pre-clinical data on its potential efficacy. We are planning a separate papers on the patient-derived GBM cell data.
The only kind of data presented here is qRT-PCR. To confirm these results as well as to investigate the possible explanations that the authors come up with for their observations, protein expression (western blot) as well as functional assays, ion channel/intracellular ion concentration determination assays and exploration of possible signaling pathways would be needed, none of which has been done.
Answer
The data we have provided are from the indicated project. The cell lines cells data have been used as a basis for studies on patient-derived GBM cells in the same project. We agree that it would be important to support the data with other data (Cl- transport assays, intracellular Cl- concentration measurement, carrier protein (Western blot) assays), but this would be a future research project based on the our findings. We have experience in intracellular Cl- concentration and anion transport assays (DOI: 10.1155/2014/569650; DOI: 10.1152/ajprenal.2001.280.2.F314). We have therefore indicated in the “Discussion” section that the data presented in the paper have limitations and that this is a challenge for future research projects. We thank the reviewer for his suggestion.
In the introduction, the authors say that VPA suppressed NKCC1 expression in thymocytes, but it seems to increase NKCC1 expression in GBM cells. What is the (possible) explanation for this?
Answer
We cannot yet answer why there are differences in the effect of VPA on NKCC1 expression when comparing data from rat (doi: 10.1177/1559325819852444) and GBM cell lines cell. This may be related to the species of animal or different origin of tissue. We are doing further studies in a different animal species, so we do not have an answer to the question at the moment.

Reviewer 2 Report
The article "Different Effects of Valproic Acid on Pediatric Glioblastoma PBT24 and SF8628 Cells as a Key to Personalisation of Therapy" describes the potential administration of valproic acid in the treatment of glioblastoma, for two sex-separate lines.
The English language needs some minor polishing for style and typos.
In introduction a stronger recent literature survey is a must, especially on recent literature. Antitumoral personalisation therapies should be shortly presented. The author need to update the introduction part by following doi: 10.3390/pharmaceutics13091356 and doi: 10.2174/1871520618666181109112655.
Authors should better explain the motivation behind chosen this treatment, in the literature context. An extensive comparison with the literature results must be done. How is this system a better one? What is the improvement?
Conclusion section must be reworked to underline the novelty and novel advances of this research.
Author Response
Response to 2nd Reviewer
English language and style
(x) English language and style are fine/minor spell check required
Can be improved
Answer
English revised and corrected
Does the introduction provide sufficient background and include all relevant?
Can be improved
Answer
We have updated the Introduction in line with comment. We have also added a figure which we think explains the design of the study. We hope that the corrections are consistent with the reviewer's comment.
Is the research design appropriate?
In light of the reviewers' comments, we have added Figure 1 in the Introduction section. We have also indicated the limitations of the study in the Discussion. We hope that the study design is now more visible and understandable.
Are the conclusions supported by the results?
Can be improved
Answer
The conclusions have been adjusted in line with the comment. Thank you for your valuable comment.
Glioblastoma PBT24 and SF8628 Cells as a Key to Personalisation of Therapy" describes the potential administration of valproic acid in the treatment of glioblastoma, for two sex-separate lines.
The English language needs some minor polishing for style and typos.
Answer
English revised and corrected
In introduction a stronger recent literature survey is a must, especially on recent literature. Antitumoral personalisation therapies should be shortly presented. The author need to update the introduction part by following doi: 10.3390/pharmaceutics13091356 and doi: 10.2174/1871520618666181109112655.
Answer
Thanks to the reviewer for his comment. We have made changes to reflect it, citing more recent references. We have also used and cited quoted one of the references.
Authors should better explain the motivation behind chosen this treatment, in the literature context. An extensive comparison with the literature results must be done. How is this system a better one? What is the improvement?
Answer
In line with the reviewer's comment, we have added Figure 1 in the Introduction section, which we hope will give a better understanding of the design and purpose of our study. We thank the reviewer for his comment. We hope that the changes made have improved the manuscript.
Conclusion section must be reworked to underline the novelty and novel advances of this research.
Answer
Thank you for your comment. We have made changes to the conclusions. We hope that we have fulfilled the requirement.

Round 2
Reviewer 1 Report
The authors have not adequately addressed my major concerns and have not performed additional experiments or included additional data to support their claims. The data remain quite preliminary and are inadequate to make any general conclusions and thus have questionable impact on the field.
Author Response
Reply to Reviewer’s comments
Comments and Suggestions for Authors
The authors have not adequately addressed my major concerns and have not performed additional experiments or included additional data to support their claims. The data remain quite preliminary and are inadequate to make any general conclusions and thus have questionable impact on the field.
Thanks to the reviewer for his repeated comments. We have carefully re-evaluated everything and have made some changes - additions: a change of the title, additions to the text of the Discussion. We hope that we have now at least partially solved the problem. Please find below our explanations.
Answers
- Reviewer pointed out the important common point that protein activity is determined by multiple mechanisms, not just gene expression (mRNA).
We mentioned in the Discussion section that this is a limitation of our study, where only mRNA expression is analysed and possible mechanisms are discussed. However, even in prognostic and therapeutic models, the analysis of mRNAs in a single method to analyse the effect on methylation, treatment effectiveness, prognosis, personalised therapy is applied in many studies (1, 2, 3, 4).
(1) Chen Z, Liu B, Yi M, Qiu H, Yuan X. A Prognostic Nomogram Model Based on mRNA Expression of DNA Methylation-Driven Genes for Gastric Cancer. Front Oncol. 2020 Nov 24;10:584733. doi: 10.3389/fonc.2020.584733.
(2) Deng GC, Sun DC, Zhou Q, Lv Y, Yan H, Han QL, Dai GH. Identification of DNA methylation-driven genes and construction of a nomogram to predict overall survival in pancreatic cancer. BMC Genomics. 2021 Nov 3;22(1):791. doi: 10.1186/s12864-021-08097-w.
(3) Park JY, Zheng W, Kim D, Cheng JQ, Kumar N, Ahmad N, Pow-Sang J. Candidate tumor suppressor gene SLC5A8 is frequently down-regulated by promoter hypermethylation in prostate tumor. Cancer Detect Prev. 2007;31(5):359-65. doi: 10.1016/j.cdp.2007.09.002.
(4) Davey MG, Cleere EF, O'Donnell JP, Gaisor S, Lowery AJ, Kerin MJ. Value of the 21-gene expression assay in predicting locoregional recurrence rates in estrogen receptor-positive breast cancer: a systematic review and network meta-analysis. Breast Cancer Res Treat. 2022 Apr 15. doi: 10.1007/s10549-022-06580-w.
We have changed the title of the manuscript to reflect the reviewer's comment. The new title is: “Different Effects of Valproic Acid on SLC12A2, SLC12A5 and SLC5A8 Gene Expression in Pediatric Glioblastoma Cells as an Approach to Personalised Therapy”
- The reviewer's comment states that the manuscript presents only qRT-PCR data, that our data are uninformative and cannot be generalised, and that protein expression (Western blot) data are needed to verify our results.
The authors would like to point out the following in response to the comment:
1) The likelihood that short-term VPA treatment would be associated with changes of protein expression is questionable. The several-fold increase in the expression of SLC12A5 and SLC5A8 after VPA treatment is important and directly reflects the correlation with methylation, and detected differences between the compared GB cells may also be therapeutically important. Researcher indicate that the mRNR analysis is applied for more quantitative analysis of gene expression, and the assay is more precise gene activity identifier comparing DNA methylation (5). Therefore, we believe that evidences of clear changes in gene expression obtained after VPA treatment are significant in itself. It is therefore questionable whether the absence of a link between gene expression and protein expression would negate the significance of our data.
(5) Phillips T. The Role of Methylation in Gene Nature Education 2008; 1(1):116. Expression (Write Science Right) © 2008 Nature Education)
2) Authors would like to point out that according to the scientific literature the correlation between differentially expressed mRNA and mRNA/protein of the same gene is debatable. Genome-wide correlation between expression levels of mRNA and protein are notoriously poor; correlations between differentially expressed mRNA profiles were low and even negative (6, 7, 8).
(6) de Sousa Abreu, R., Penalva, L. O., Marcotte, E. M. & Vogel, C. Global signatures of protein and mRNA expression levels. Mol. Biosyst. 2009; 5, 1512–1526.
(7) Vogel, C. & Marcotte, E. M. Insights into the regulation of protein abundance from proteomic and transcriptomic analyses. Nat. Rev. Gen. 2012; 13, 227–232 ().
(8) Koussounadis, A., Langdon, S., Um, I. et al. Relationship between differentially expressed mRNA and mRNA-protein correlations in a xenograft model system. Sci Rep 2015; 5, 10775. https://doi.org/10.1038/srep10775
- The reviewer pointed out that functional studies, such as ion channel and/or intracellular ion concentration studies, have not been carried out to investigate possible signal transduction pathways.
The authors would like to point out the following in response to the remarks:
1) A limitation of our study was that other mechanisms underlying protein activity were not excluded. Our data include factors that determine Cl- influx (NKCC1) and anion efflux from the cell (KCC2 and its associated GABAA receptor). Therefore, in our opinion, it would be important to study the effect of VPA on intracellular Cl- concentration ([Cl-]i), which itself is a signalling pathway, and because the [Cl-]i level is a common result of all mechanisms regulating Cl- transport. At the same time, we have to say that we cannot add data from such studies to the manuscript, as this would be a separate project that would require separate funding.
2) The requirement to provide data on Cl- channels study is, in our opinion, redundant, as this is a different line of research, which is beyond the scope of our project.
To explain the limitations of the study, we have added additional text to the Discussion section.
Reviewer's comment on GB patient-derived cell studies
In our opinion, this is a redundant requirement as it is a task for a separate study.
- Regarding the comment on conclusions
Our findings are not intended to be generalised, but rather to provide a comparison of two cell lines that allows us to draw conclusions about the individual treatment approach. By changing the title of the manuscript, we believe that the conclusions also adequately reflect the results obtained.

Reviewer 2 Report
The authors have responded to my comments and have addressed all my concerns, substantially improving the manuscript, therefore, I suggest publishing the paper in the current form.
Author Response
The authors have responded to my comments and have addressed all my concerns, substantially improving the manuscript, therefore, I suggest publishing the paper in the current form.
Answer
The authors sincerely thank the Reviewer for his positive evaluation of the manuscript.